# Twenty Years of Mortgage Banking in Slovakia

**Eva Horvatova** [1,2]

[1]   Faculty of National Economy, University of Economics in Bratislava, Dolnozemská 1,
852 35 Bratislava, Slovakia; eva.horvatova@econ.muni.cz or eva.horvatova@euba.sk
[2]   Faculty of Business Administration, Masaryk University Brno, Lipova 41 a, 601 77 Brno, Czech Republic

**Abstract:** Mortgage banking began to develop in Slovakia after 1998 as an ambitious project, the goal of which was to elevate the lagging development of the real estate market, the development of the financial market and the creation of banks' long-term resources. Our goal is a comprehensive assessment of the development of Slovak mortgage banking for the past 20 years from the perspectives of the development of banking, the mortgage bond market, the real estate market and selected interactions between individual elements of the mortgage system. The specific aim of the study is to evaluate the substantial links between the basic economic indicators, indicators of housing finance and real estate prices in Slovakia. To evaluate these issues VAR (Vector Autoregression) models, models of panel and linear regression and DEA (Data Envelopment Analysis) models were used. Slovakia has specific indicators of the development of mortgage banking, adequate to its historical and economic development. It was confirmed that the availability of real estate loans had a significant impact on the increase in real estate prices. Real estate prices in Bratislava have different development factors than real estate prices from a nationwide perspective. Low interest rates have an important role in housing financing. The second part of the study is oriented towards an evaluation of the technical efficiency of individual banks. The results of DEA point out that the largest banks in Slovakia were the most efficient in the pre-crisis year 2007. The overall results show that policymakers should react not only to the household indebtedness rate and risks for individual clients, but should also see the risks for banks in possible changes in the real estate market, or the risks of changes in interest rates in the future.

**Keywords:** housing credits; interest rates; real estate prices

**JEL Classification:** G21; G28; R31

## 1. Introduction

The term "mortgage system" describes an integrated system that begins with the procurement of resources, followed by offering these resources in the form of mortgage loans, their repayment by the mortgage debtors and, subsequently, the mortgage banks paying the resources back to their providers. In order for this system to work, it has to be sustainable and stable. The system of rules of mortgage banking focuses primarily on the mortgage banks' ability to pay their liabilities to their creditors, i.e., other subjects that purchased securities in the form of mortgage bonds.

Mortgage banking has features that are common for mortgage systems in all countries, as well as particularities that are different in individual countries. If we see a significant level of integration and convergence of markets in the field of banking and finance, the integration of mortgage markets takes place very slowly and is caused by the fact that there are significant differences among individual countries in the price of real estate and in the field of real estate management legislation. The integration of financial markets is slowest in the field of real estate markets.

From an international comparison point of view, mortgage systems have common characteristic features that are evident in every system, but also specific features that are subject to specific historic development in a given state.

Basic features of mortgage systems are: strict conditions for entry in the sector, protection of investors (owners of mortgage bonds), measures focused on the stability of mortgage banks, coverage of mortgage bonds, i.e., covering the resources of the mortgage bank by the assets of the mortgage bank, strict monitoring of the purpose of using the resources of the mortgage banks (i.e., for mortgage loans, with the exception of assets that serve alternate coverage), and alternate coverage limited to a specific limit (usually 10–20% of the value of the issued mortgage bonds).

The issuance and sale of mortgage bonds are crucial in mortgage banking. They represent investment opportunities for different investors in the banking and financial markets. For mortgage bonds to be a reliable investment, they have to follow strict rules regarding the management of the acquired resources.

The protection of investor interests, i.e., the owner of the mortgage bonds, is a major principle of mortgage banking. Previous ideas about the need for stability of mortgage banking led to the definition of a protection system in the form of the basic rules of mortgage banking. Therefore, the quality of issued securities used to finance loans is very important. One of the major themes of the European financial sector is the area of harmonization of covered bond regulations.

Mortgage banking has historically developed as an area of banking focused on financing real estate in the private, public and state sectors. It is a strong factor in the development of bank credit activities, it contributes to the development and stability of financial markets and it has a positive effect on the development of the economy. At the same time, the mortgage system is sensitive to macroeconomic conditions and bank risks. Mortgage banking is based on long-term relationships, because real investment and real estate construction have a long-term nature. This gives rise to additional risks on the side of banks and creditors, as well as from the conditions of market development.

The existing literature contains many new insights into housing finance and mortgage banking and for evaluating its development in Slovakia. However, there is a lack of studies looking at systemic issues regarding the development of mortgage financing in Slovakia. Some results of this research may be relevant and useful for comparison with other CEE countries because they have similarly developed mortgage systems.

This research is motivated by the importance of housing finance and housing credits for the banking sector in Slovakia; however, the importance of this issue is much broader, for example, for real estate markets, and for the regulation of housing finance due to LTV (Loan to Value Ratio), PTI (Payment to Income Ratio) and DTI (Debt to Income Ratio).

The paper is important in two respects: (1) it investigates factors which have affected the housing finance and mortgage banking in Slovakia at the banking sector level; and (2) considering that the largest banks in the Slovakian banking sector belong to Western European banks (Erste Bank, Intesa Sanpaolo, Raiffeisen, UniCredit), and that the stability of these banks may also affect banks in Western Europe, the paper analyzes indicators at the individual bank level.

The paper is structured as follows. The first aim of the study is to evaluate the substantial links between basic economic indicators, indicators of housing finance and real estate prices in Slovakia. To evaluate these issues we used VaR models, and models of panel and linear regression. It can be assumed that individual countries have their own specific indicators of mortgage financing.

The second part of the analysis focuses on the evaluation of the technical efficiency of individual banks using the indicators of housing finance and mortgage banking. Measuring the efficiency of banks was carried out using Data Envelopment Analysis (DEA) models.

## 2. Theoretical and Methodological Basis of Mortgage Market Research

Mortgage systems are divided into these basic types: 1. specialized mortgage banking, in which there are specialized institutions providing exclusively mortgage transactions; 2. mixed type, in which

the banks provide mortgage transactions with additional services; and 3. universal, in which banks provide all types of banking services (Pavelka 1996).

Historically, there were three basic models of mortgage banking (Sivák et al. 2007): (1) the deposit model, based on providing of mortgage loans from primary deposits in banks; (2) the model based on issuing mortgage bonds, in which banks provide mortgage loans in this manner; and (3) the model based on securitization, i.e., on the existence of the primary and the secondary mortgage market. The third model is characterized by pooling of mortgage loans into homogenous groups called pools, which are subsequently sold to subjects that use them as collateral for the issuing of mortgage bonds, through which they acquire new resources for providing mortgage loans. This model is used mainly in the U.S. The securitization of mortgage assets is the result of the natural evolution of a developed market in stable economic conditions. It requires large-scale trading and the entities performing the securitization of assets having investor confidence and high rating.

Mortgage banking as a system that connects the banking, the financial and the real estate markets is a subject which can be examined from many different perspectives. Blackwell and Kohl (2019) have analysed the historic conditions of the creation of mortgage systems and their effect on industrial development. They divide historical housing systems into three groups: 1. direct finances, using direct ways of financing, not mediated by banks or state institutions; 2. deposit based finances, using customer deposits to provide mortgage credit; and 3. bond based finances, which rely on the sale of mortgage bonds to finance mortgage credit. The analysis of mortgage markets covers several aspects. Overall, they can be divided based on their impact on banks, real estate markets, bond markets and portfolios of investments.

In terms of geographical breakdown, according to the authors Blackwell and Kohl (2017), the majority of OECD countries developed along two main trajectories of urban housing finance during the nineteenth century: one based on the collection of deposits, and the other based on the sale of bonds on capital markets. They found that Anglo-Saxon and North-Western European countries developed a predominantly deposit-based system of housing finance, whereas central continental European and Scandinavian countries developed bond-based systems in order to finance their urban expansion, with varying degrees of state sponsorship. While the Southern and Eastern European countries upheld the direct finance model much longer than their central and northern counterparts, these countries generally developed deposit based systems as the twentieth century progressed.

According to Sunega and Lux (2007), several authors agree that "the differences arise from local housing finance traditions, macro-economic performance, accessibility of funds, sources of capital for loan extensions, variety and types of mortgage products, variety of interest rate fixing, and level and content of state interventions".

The issue of mortgage bond regulation is addressed by regulatory and supervisory authorities that require standards for issuing covered bonds with a high level of investor protection. Since there are several versions of mortgage securities, such as mortgage bonds, mortgage-backed bonds, or covered bonds, the issue should be examined from different angles regarding which are the most profitable for investors and banks. Carbó-Valverde et al. (2017) have addressed this issue. The structure of mortgage bond securities is not the same as the structure of covered bonds. The main difference is that, with mortgage-backed securities, more risk is transferred from the banks to the bondholders than with covered bonds. "Examining the use of covered bonds (CB) and mortgage backed securities (MBS) in the U.S. and Europe, we find that the two often seem to be used for different purposes. Banks are more likely to use covered bonds when they have liquidity needs, while MBS are associated with risk management and agency problems. Introducing MBS to markets where only CB are common or CB to markets where only MBS are common could have large effects."

Carbo-Valverde et al. (2015) also examined the impact of household financial constraints on the mortgage rate from different perspectives of credit risk. Their research focused on the Spanish market in the years 2002, 2005, 2008 and 2011. Their results show that the risk profile does not have a

significant effect on mortgage rates and that, during the financial crisis, the credit institutions tended to have higher rates that did not take into consideration the risk profile of the customers.

Vukovic (2015) analyzed real estate portfolio management in banks: how real estate prices reacted to shocks, using VAR and SVAR (Structural Vector Autoregression) models, Mahalanobis distance and impulse response functions. This paper analyzed real estate markets in Austria, Germany and Switzerland.

Given the known stabilization effects of mortgage banking on the economy, it is interesting to analyze its impact on the level of financial stability. Morgan and Zhang (2017) examined the relationship between financial stability and mortgage lending from several perspectives. They examined the effect of the share of mortgage lending by individual banks on two measures of financial stability—the bank Z-score and the non-performing loan ratio—on a sample of 1889 banks in 65 developed and emerging economies during the period 1987–2014. The authors compared the behavior of banks in Europe and Asia and found a positive relationship between increased shares of mortgage lending and financial stability, specifically by lowering the probability of default by financial institutions and reducing the non-performing loan ratio, at least in non-crisis periods, for levels of mortgage shares up to 49–68%. The impact on financial stability was negative for a higher level of mortgage lending shares. Compared with the base sample, Asian banks had better financial stability during non-crisis periods, but they were more negatively affected by a higher mortgage ratio during crisis periods, while European banks had greater financial stability during non-crisis periods, but they were less negatively affected during crisis periods. A higher level of regulatory quality improves financial stability in both groups of countries.

A paper by Dröes and Francke (2018) examined which factors determined the correlation between prices and turnover in European real estate markets. They used the panel vector autoregressive model and they discovered a strong feedback mechanism between prices and turnover. Momentum effects are the primary reason why prices and turnover are correlated. Macroeconomic factors, such as GDP and interest rates, could also explain part of the price—turnover correlation. The results of the paper imply that it is important to model prices and turnover as two interdependent processes to understand price and turnover dynamics.

Kelly et al. (2018) evaluated Irish mortgage credits originated during the period 2003–2010 and constructed a model of credit availability as a function of income, wealth, age, interest rates and market conditions. The house price model shows that a 10% increase in credit availability leads to a 1.5% increase in the value of property. Coefficients from the model were used to fit values under scenarios of macroprudential restrictions on LTV (Loan to Value ratio), PTI (Payment to Income ratio) and DSR (Debt Service Ratio) on credit availability and house prices in Ireland during the 2003 to 2010 period. Their results suggest that market conditions such as LTV, PTI and DSR have had significant impact on house prices.

Greenwald (2018) found in his research that Payment to Income Ratio (PTI) is a more effective tool to dampen property price growth than Loan to Value Ratio (LTV).

The paper of Černohorský (2017) is one of the studies examining the effects of loans on the economy. The author examined the effect of different types of bank loans on economic development. He assumed that economic performance increases with the growth of various types of loan. Results of the analysis confirmed that different types of bank credits have significant impact on the economy. Consumer loans were the only exceptions where this relationship was not confirmed. Other types of loans—loans to non-financial businesses, loans to households, mortgage loans and total loans—influence the development of gross domestic product.

The issue of choosing between a fixed and a variable interest rate for mortgage loans is topical in the context of the risks of mortgage banking. Magdoň (2018) deals with this topic using the example of Poland, where almost 100% of all mortgages are provided with a variable interest rate. In his paper he defends the introduction of fixed interest rates as a tool for protecting creditors against changes in interest rate. In this context it can be said that mortgage banks in Slovakia provide the majority of their mortgage loans with interest rates fixed for a defined period of time (3 years, 5 years, or 10 years).

In this context it is necessary to take into consideration other aspects, such as the motivation of the debtors to prepay their mortgage loans. Groot and Lejour (2018) analyzed heterogeneous responses in prepayment behaviour: "because interest rates on savings have decreased and the rates on existing mortgages are generally fixed for some years, the value of prepaying a mortgage has increased." The authors find a statistically significant effect on prepaying mortgage credit of the interest differential (e.g., the difference in the interest rate that households pay on their mortgage minus the interest rate they receive on their savings account). Therefore, it is necessary to take into consideration not only the amount and type of interest rate on the mortgage loan, but also the interest rates on the resources of banks and clients.

Mortgage banking is highly sensitive to macroeconomic development conditions. Zhang and Xu (2018) suggest that monetary policy supports the securitization of assets, the issuance of MBS (Mortgage Backed Securities), which is driven by the transfer of risks to other entities. They show that "low interest rate not only directly boomed the mortgage lending and softened banks' lending standards but also was an important motive for banks to securitize. Massive financial innovations were not the cause of the housing bubble but a rational response of banks to ease monetary policy."

Another view on mortgage market securitization offers the aspect of financialization of the economy. Schwartz (2020) describes the (re-)nationalization of housing finance in the context of increasing securitization: "because real estate debt cannot be transformed into tradable assets without securitization, financialization requires state covering for the MBS market in particular and asset-backed securities more generally. As in C19, the state is the motor behind expansion of this market."

In the United States, compared to other mortgage systems, the prepayment of mortgage loans is unlimited. Therefore, it is necessary to estimate what percentage of loans will be repaid in order to subsequently repay mortgage-backed securities.

Joseph C. Hu (2011) states: "to measure the paydown of a mortgage pool with respect to its age, the Public Securities Association (now the Securities Industry and Financial Markets Association, SIFMA) promulgated a new yardstick termed percent PSA. (Interestingly, the PSA has also been interpreted as being short for Prepayment Speed Assumption.) The 100% PSA refers to a pool that prepays at a 0.2 percent CPR (Conditional Prepayment Rate) per month during the first 30 months, and after at a constant 6 percent CPR per month at the 31st month and thereafter."

Sicakova-Beblava and Beblavy (2016) investigated the impact of state regulation on the housing loan market during the years 2010–2015 in Slovakia. This issue was examined on the basis of changes in economic policy. Their research shows that an increase in competition in the housing loans market leads to a significant decrease in interest rates.

Mortgage banking is closely linked to the regulation of banking. Uluc and Wieladek (2018) have studied the effect of changes in bank capital requirements on mortgage loans during the period 2005Q2–2007Q2 in the UK. They found that an increase of 100 basis points in capital requirements leads to a 4% decline in the size of an individual mortgage. This leads to the growth of riskier loans in respect to growth of capital.

Repullo and Saurina (2011) found that, applying a countercyclical capital cushion, banks preferred to restrict lending. This would mean that the countercyclical capital buffer would become a pro-cyclical instrument. Increasing capital and reducing credit are alternative solutions.

The stable real estate market is the most important condition for the stability of the banking system. Several authors deal with the question of mortgage banking in relation to prices of real estate and to the creation of a speculative bubble.

An interesting methodological approach to the analysis of the real estate market was chosen by Brissimis and Vlassopoulos (2009). They also involved a time aspect in their analysis: short-term and long-term views. The aim of their paper was the analysis of the interaction between housing loans and housing prices in Greece using multivariate cointegration techniques. The results of their long-running analysis indicated that "housing prices are weakly exogenous, hence they do not react to disequilibria in the mortgage lending market. This suggests that in the long run a line of causality running from

housing loans to housing prices is not confirmed". The short-run analysis indicated a bi-directional dependence among housing loans and housing prices.

The mortgage effect is a sequence of successive events that begins with rising property prices, pushing consumers to buy real estate, and this causes a bubble in the market. For companies that own real estate, the value of collateral for loans increases, and their potential to draw new loans increases. "The mortgage effect enhances the bargaining power of enterprises and makes it easier for enterprises to obtain external financing." (Li and Gao 2019). The authors state that "the mortgage effect does exist and the authors further analyze the heterogeneity of this effect by dividing the sample based on the degree of financial development and property rights; the empirical results reveal that the mortgage effect is significantly higher in countries with high level of financial development. Besides, compared to the SOE (State-owned enterprise), the mortgage effect has more influence on non-SOE companies." This paper makes an important finding for policymakers, that "regulating the housing market, the government should take impact on corporate financing . . . and that although the mortgage effect has a positive effect on the financing ability of enterprises, it only exists in the short term and is not sustainable".

Wu and Lux (2018) analyzed U. K. regional real estate prices from 2005 to 2017 to identify factors influencing house prices (regional versus national) and potential price bubbles. They used the Gordon dividend discount model and they considered house prices as the present value of imputed rents. They differentiated between long-term and short-term effect using pooled mean group (PMG) and mean group estimation (MG) to determine variations in regional house prices. Regional trend analysis shows that house price growth in the regions has been affected differently in the short run and each region has varying long-run fundamentals.

Brauner and Plottová (2017) deal with the opposite relationship—they investigated which factors affect the price of rent. They also used Gordon's growth model for this purpose. They found that the price of rent depends on the purchase price of property, which is influenced by the purchasing power of the population, by interest rates, by development of the financial market and by the inflation rate.

It is obvious that there is an interdependent relationship between property prices and rental prices. Cronin and McQuinn (2016) found that a reduction in the LTV (Loan-to-Value Ratio), as a result of regulatory limits, will lead to growing demand for rental accommodation prompting higher rents for a given house price level.

In the countries of Central and Eastern Europe, the share of mortgage debt to GDP is rising and is accompanied by a rise in house prices. There is a high degree of home ownership, which originated without a market environment.

Bohle (2018) argues that the EU framework for free movement of capital and financial service provision as well as the availability of cheap credit has induced a trajectory of housing financialization, which has two forms: (1) funding from wholesale markets; and (2) direct penetration of foreign financial institutions. "These two forms attest to a core-periphery relationship in housing financialization, whose hierarchical character came to the fore in the crisis." In all four analysed countries (Iceland, Ireland, Hungary and Latvia) EU-induced external liberalization, domestic policies of deregulation and privatization of the banking sector have channeled predominantly international liquidity into mortgage finance.

Property has a key role in mortgage banking: it is the subject of financing and at the same time serves as collateral for credit protection. The fall in the real estate market represents the most significant risk of mortgage banking, as has also been shown during the US mortgage crisis. Clients were not able or willing to repay loans because cheaper properties were available on the market. The problem of falling property prices has fully impacted the banks.

Vogel and Werner (2015) analyzed bubbles and found that extreme market events are principally caused by excessive bank credit extensions for leveraged transactions that ignite and propel unsustainable market movements. "With the further development of volatility metrics such as those proposed, central bankers and governments might begin to assess economic and financial

market conditions from new perspectives. This should enable better-informed policy formulation and, hopefully, mitigation of the most pernicious aspects of bubbles and crashes."

Recent issues in the development of the mortgage market also include multiple property ownership for different purposes—for own living, for rent, for sale, as well as what impact these forms of ownership have on the real estate market. Huang et al. (2020) deal with this question using the example of China. It would be appropriate too to analyze the reasons for and impact of multiple property ownership in European countries, because this can be an important factor in house price growth.

## 3. Methodology and Data

### 3.1. Empirical Methods

In the article, the VAR model was used to determine the responses to the selected impulses in mortgage banking in Slovakia. The VAR model is a linear interdependent model with more than one dependent variable. The basic form of the VAR model consists of a set of K endogenous variables (Pfaff 2008).

$$y_t = (y_{1t}, \dots y_{kt}, \dots y_{Kt}), \tag{1}$$

for $k = 1, \dots K$.

The VAR(p) process is then defined as:

$$y_t = A_1 y_{t-1} + \dots + A_p y_{t-p} + u_t, \tag{2}$$

where $A_i$ are $(K \times K)$ coefficient matrices for $i = 1, \dots, p$ and $u_t$ is a K-dimensional process with mean value $E(u_t) = 0$ and time invariant positive definite covariance matrix $E(u_t, u_t^T > t) = \Sigma u$ (white noise); (Pfaff 2008).

For the impulse-response functions, it is necessary to use the presented reduced form of VAR (without contemporaneous terms) to identify the effect of the shock on the dependent variable. As Vukovic (2015) states, Cholesky decomposition is one of the leading factors in reducing VAR models using lower- or upper- triangular matrix. Cholesky decomposition ensures that the residual covariance matrix could be transformed into a diagonal matrix. The purpose of this Cholesky decomposition is that the impulse to one variable (or associated innovations) must be unrelated to the impulse in another variable, otherwise it is unrealistic to assume that one variable would remain static (no impulse) while the other moves (impulse).

This paper also uses the panel regression model. Firstly, this model was used to determine which factors affect the indicators of mortgage banking in Slovakia. The credit and real estate prices were dependent variables. The data in the panel is presented in Table 1. The basic form of the panel regression model is expressed as:

$$y_{it} = \alpha_{it} + \beta_{it}^T - x_{it} + u_{it} \tag{3}$$

where:

$i = 1, \dots, N$ = sectional index
$t = 1, \dots, T$ = time index,
$u_{it}$ = random error
$\alpha$ = omitted effects.

The Hausman test is used to select one of the two models (fixed or random).

The DEA (Data Envelopment Analysis) model is used to analyze the impact of bank mortgage business indicators on the technical efficiency of banks in Slovakia. DEA analysis is one of the non-parametric methods of measuring efficiency. The disadvantage of the DEA model is that it is not possible to separate the effect of random errors and errors in the measurement of inefficiency. DEA measures the relative efficiency of production units in the examined group of units. By changing the group, we can expect a change of efficiency in the examined units. M. J. Farrell was one of the

first scientists to develop the DEA model. His best known work is "The Measurement of Productive Efficiency" published in the Journal of the Royal Statistics Society in 1957. In this paper, we have also the used input-oriented Charnes-Cooper-Rhodes-Input (CCR) model and Banker-Charnes-Cooper-Input (BCC) model.

The CCR model can be written in the form of a linear programming problem (Jablonský and Dlouhý 2004):

$$\max z = \sum_i^r u_i \cdot y_{iq} \tag{4}$$

Under these conditions:

$$\sum_i^r u_i \cdot y_{ik} - \sum_j^m v_j \cdot x_{jk} \leq 0; \ k = 1, 2, \ldots n \tag{5}$$

$$\sum_j^m v_j \cdot x_{jq} = 1 \tag{6}$$

$$u_i \geq \varepsilon \quad i = 1, 2, \ldots r$$

$$v_j \geq \varepsilon \quad j = 1, 2, \ldots m$$

The input-oriented BCC-I model can be written in the form (Jablonský and Dlouhý 2004).

$$\max z = \sum_i^r u_i \cdot y_{iq} + \mu \tag{7}$$

Under these conditions:

$$\sum_i^r u_i \cdot y_{ik} + \mu \leq \sum_j^m v_j \cdot x_{jk} \ k = 1, 2, \ldots n \tag{8}$$

$$\sum_j^m v_j \cdot x_{jq} = 1 \tag{9}$$

$$u_i \geq \varepsilon \quad i = 1, 2, \ldots r$$

$$v_j \geq \varepsilon \quad j = 1, 2, \ldots m$$

$$\mu = optional$$

The $\mu$ parameter reflects the conditions of convexity of the BCC-I model

### 3.2. Sample and Variables

The data have been sourced from the National Bank of Slovakia (Central Bank of the Slovak Republic), as well from financial statements of the banks in Slovakia.

Used variables are in Tables 1 and 2:

PROVIDED LOANS = Provided housing loans (credprow)
NEW CREDITS = New housing loans (crednew)
MORTGAGE BONDS = Mortgage bonds (bondsemit)
RESIDENTIAL PROPERTY PRICES = Residential property prices as average prices for Slovakia
BRATISLAVA = Real estate prices for region Bratislava
TRNAVA = Real estate prices for region Trnava
NITRA = Real estate prices for region Nitra
TRENCIN = Real estate prices for region Trencin
ZILINA = Real estate prices for region Zilina
BANSKA BYSTRICA = Real estate prices for region Banska Bystrica
KOSICE = Real estate prices for region Kosice
PRESOV = Real estate prices for region Presov
D_HDP = Change of GDP
D_INFLATION = Change of inflation

UNEMPLOYEMENT = Unemployment rate

D_WAGES = Changes of wages

IR_CRED_ONTOFIVE = Interest rates of credits with maturity one–five years

IR_CRED_OVERFIVE = Interest rates of credits with maturity over five years

LONGTERM_IR_DEBT_SEC_TEN_Y = Long-term interest rates of debt securities over ten years.

**Table 1.** Variables and descriptive statistics of Slovak mortgage banking development during the period 2006–2017.

| Variable | Minimum | Maximum | Mean | Std. Dev. |
|---|---|---|---|---|
| PROVIDED LOANS | 2,261,761 | 97,983,32 | 5,990,776.5 | 1,954,713.7 |
| NEW CREDITS | 176,492 | 1,087,826 | 719,341.5 | 271,918.16 |
| MORTGAGE BONDS | 1,513,056 | 4,952,103 | 3,481,777.9 | 777,605.048 |
| RESIDENTIAL PROPERTY PRICES | 944 | 1549 | 1262.20 | 120.792 |
| BRATISLAVA | 1292 | 2019 | 1703.99 | 147.033 |
| TRNAVA | 707 | 1055 | 850.56 | 76.443 |
| NITRA | 370 | 796 | 596.76 | 94.214 |
| TRENCIN | 405 | 867 | 665.16 | 90.003 |
| ZILINA | 488 | 973 | 773.93 | 107.724 |
| BANSKA BYSTRICA | 490 | 866 | 733.78 | 88.534 |
| KOSICE | 536 | 1186 | 919.59 | 133.173 |
| PRESOV | 581 | 1157 | 805.64 | 105.872 |
| D_HDP | −0.41 | 0.88 | 0.3149 | 0.31871 |
| D_INFLATION | −0.040 | 0.360 | 0.13246 | 0.142394 |
| UNEMPLOYEMENT | 9.6 | 16.2 | 12.491 | 2.0615 |
| D_WAGES | 0.12 | 0.36 | 0.2489 | 0.06634 |
| IR_CRED_ONTOFIVE | 1.6774 | 6.5000 | 4.558701 | 1.4376583 |
| IR_CRED_OVERFIVE | 1.6313 | 9.0900 | 6.236752 | 1.7341660 |
| LONGTERM_IR_DEBT_SEC_TEN_Y | 0.3010 | 5.8000 | 3.333270 | 1.6229458 |

Source: National Bank of Slovakia 2018.

**Table 2.** Variables used for the panel regression models and for DEA models were.

| | |
|---|---|
| HOUS_CRED | = Housing credits (Mortgage credits + Other credits); |
| MTG_CRED | = Mortgage credits (Special credits defined in the Banking Act); |
| OTHER_CRED | = Other housing credits; |
| MTG_BONDS | = Mortgage bonds; |
| DEPOS | = Customers deposits; |
| PROFIT | = Profit of banks; |
| IM | = Interest Margin (Interest incomes from credits minus interest expenditures from mortgage bonds). |

## 4. Mortgage Banking in Slovakia—Stylized Facts

The Slovak banking sector has been developing under changing conditions, so the determinants of its development represent a complex of several factors. The Slovak banking system faced serious issues during the transformation of the Slovak economy after the social and economic changes in 1989. There were difficulties in acquiring long-term resources, maintaining capital adequacy requirements, and from rapid growth of defaulted loans.

The instability of the financial market in the mid-1990s was linked to the issue of high interest rates. At this time, the state was financing its deficit, which in general led to an increase in interest rates and the crowding out private investments. The largest Slovak banks had liquidity and solvency problems. These issues in the business sector and the negative factors in the economic transformation were transferred to the financial and banking sector. Given the issues of the Slovak banking sector, restructuring and subsequent privatization of state banks proved to be suitable solutions. The main goal was to get banks into a better financial state and achieve the desired ratio of capital adequacy and to significantly lower the volume of non-performing loans. The restructuring of the banking sector in Slovakia was an important stabilization factor of and a step towards the privatization of banks. Foreign

capital entered these privatized banks: Intesa BCI, an Italian banking group, entered VUB, the Austrian Erste Bank entered Slovenská sporiteľňa and the Hungarian OTP Bank entered IRB. The privatization of the largest Slovak banks led to a standard professional management of banks and to improvement in efficiency and stability. The stability of the banking sector significantly contributed to the development of mortgage banking in Slovakia, the new history of which began in 1998.

In 2004 the Slovak Republic joined the European Union. The Slovak banking sector entered a new stage of development in the common European area. Positive macroeconomic development leads to stable economic growth, direct foreign investments, increasing liquidity and falling inflation and interest rates. Lowering interest rates limited the influx of speculative capital. After joining the EU, Slovakia was perceived as a fast growing economy, which could join the Eurozone within a short period of time, by 2009. Due to the financial crisis, the banks then began to tighten up their credit policies.

Altmann (2006) saw some risk: "The fact that foreign investment in CEE banking is not very diversified among countries and banks might create additional problems for the region due to contagion rising not only from the EU-15 to the east, but also between individual CEE host countries. This is due to the fact that banks active in the regions that are experiencing losses due to a crisis, let us say in Hungary, might have to reduce exposure in the Czech Republic and thus transfer the crisis. Moreover, several CEE economies may be hit by the same contagion from one country (for instance Austria) due to the high market share of banks from a few countries in the entire region. The issue of credit stability and pro-cyclical lending behavior by foreign banks is thus a very important concern for CEE".

During the period after the crisis, the Slovak banking sector maintained its credit growth. Real estate and housing loans represent a large share of total loans. The financing of real estate was a dominant element, because the loans were used not only to purchase real estate, but also to finance development projects. Today, Slovak households are the most indebted in the EU. The credit to GDP ratio grew from 25% in 2015 to 42% in 2019.

The creation and development of mortgage banking in Slovakia was a positive element in the evolution of the banking and financial system and contributed to the development of other sectors related to investment in real estate and housing construction.

Mortgage banking in Slovakia has been developing on the basis of European historic experiences, primarily on the basis of the German mortgage system. Since the beginning, it was built as a system based on issuing mortgage-covered bonds. The aim was to build a mortgage system on solid foundations that would develop the credit market, the bond market, and the real estate market.

The required minimum capital of a mortgage bank is not uniform in individual EU countries. In Slovakia, the minimum capital for a mortgage bank was EUR 33.2 million.

There was a significant increase in the number of bank subjects in the mortgage market between 1998 and 2000. Banks that received a license during this period have kept their leading position in the Slovak mortgage market until the present. These are primarily the VÚB, a.s., Slovenska sporitelna, a.s., Tatra banka, a.s. and CSOB, a.s.

Issues in the financial market, especially high interest rates on Slovak government bonds, represented an obstacle to the development of a standard mortgage system. Several support measures represented incentives for the development of mortgage banking. These were, for example, the state contribution to the mortgage loan, or temporary exemption of profits from mortgage bonds from income tax. The stabilization of the macroeconomic indicators of the Slovak economy, the decrease of interest rates, the growth of employment, economic performance and the adoption of the Euro currency in 2009 significantly contributed to the development of mortgage banking.

The development of mortgage tools was accompanied by the development of other financial tools used to support housing construction that complemented the possibilities of financing real estate from various resources. Other products have also been developed, for example in the area of insurance and savings.

Therefore, the development of mortgage banking introduced positive synergy effects for all economic subjects. At the time, the mortgage banking system was capable of concentrating sufficient

long-term resources and guaranteed providers with adequate security and appreciation of the assets invested in mortgage bonds. Since 2017, new mortgage legislation in Slovakia only regulates covered bonds, but mortgage bonds are still catching up in practice, as some issues of mortgage bonds are valid and have not yet been paid back to the investors.

Covered bonds are a financial instrument that provide stable sources of credit for long-term financing of mortgage loans, mainly residential real estate and public sector loans, but also corporate loans and specific banking transactions. The covered bond is associated with high investor confidence and safety. This investor certainty arises from the underlying assets that secure the covered bonds and which are separated into a separate cover block (Krčmár 2017).

The Table 3 shows the chronological development of licenses issued by the National Bank of the Slovak Republic for mortgage transactions to the banks in Slovakia. There were eight mortgage banks in the Slovak mortgage market with a NBS (National Bank of Slovakia) license to conduct mortgage transactions.

**Table 3.** Banks with license for mortgage banking in Slovakia (to 31 December 2017).

| Bank Name | Start | Ending | Reason |
|---|---|---|---|
| Czechoslovakian Trade Bank (CSOB, a. s.) | 1 January 2008 | | |
| OTP Bank Slovakia (OTP Banka Slovensko, a. s.) | 13 August 2002 | | Ongoing Merger (KBC Group) |
| Prima Bank Slovakia (Prima banka, a. s.) previously Dexia | 5 August 2003 | | |
| Sberbank Slovakia (Sberbank, a. s.) | 29 July 2002 | 31 July 2017 | Merger (Prima Bank Slovakia) |
| Slovak Savings Bank (SLSP, a. s.) | 31 December 1997 | | |
| Tatra Bank (Tatra banka, a. s.) | 27 January 2000 | | |
| General Credit Bank (VUB, a. s.) | 3 June 1997 | | |
| UniCredit Bank (UniCredit Bank Czech Republic and Slovakia, a.s.) | 12 November 2013 | | |

Source: National Bank of Slovakia 2019.

The indicator of the development and significance of the mortgage market is the ratio of the mortgage debt to GDP. An extremely high ratio suggests that investment in real estate represents significant activity relative to the performance of the economy. Rapid growth can signal risks due to the excessively growing indebtedness of the population. A low share of mortgage debt in GDP does not have to mean that the development of the mortgage market is lagging, and is also found in countries with an extremely high GDP.

Traditionally, this ratio is high in countries such as Denmark, Holland, Cyprus, Norway, Spain and Portugal. Extremely low values of this indicator can be found in Albania and Romania. Between 2010 and 2017, the ratio of the mortgage debt to GDP in Slovakia ranged from 16 to 27.6%.

Countries with the highest ratio of mortgage debt to GDP include Holland, Denmark, Cyprus, Switzerland, Ireland, Spain, Portugal, Sweden, Norway, Great Britain and the U.S. A high ratio of mortgage debt to GDP represents a certain risk manifested through the real estate market, real estate price bubbles or through the credit market.

It must be noted that mortgage banking provides finance not only for residential properties but is has the potential to finance the construction of properties for the public and the state sectors, for which investment and development products are used. The segment of financing residential properties is the most developed in Slovakia.

The volume of housing loans in Slovakia has seen a continual growth since the creation of mortgage banking. There was a slight slowdown in the growth of mortgage loans during the financial crisis.

The Figure 1 shows of housing loans (provided credits and new credits), amount of issued bonds, and logarithm of provided loans.

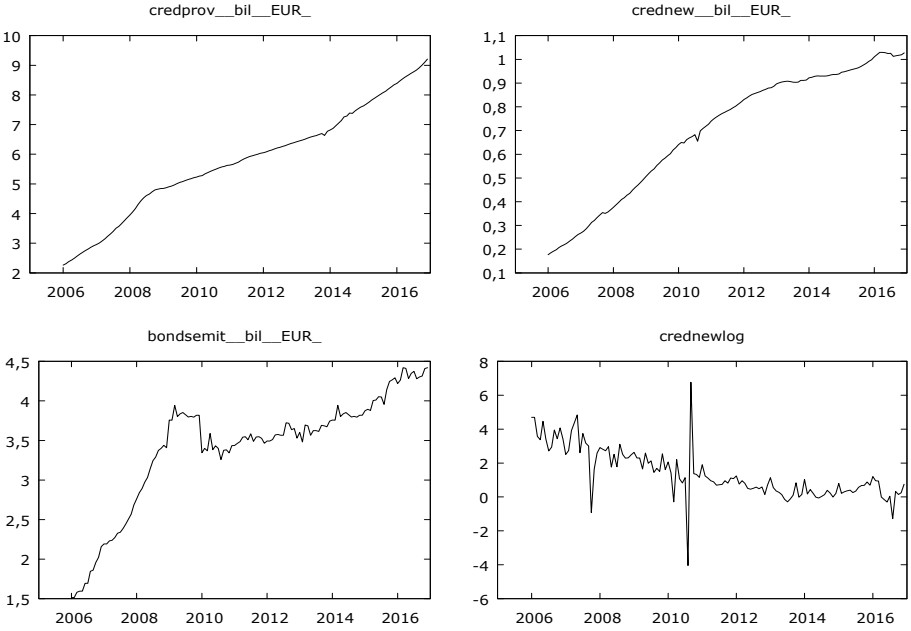

**Figure 1.** The the volume of "credprov" = provided credits, "crednew" = new credits, "bondsemit" = amount of issued bonds, "crednewlog" = logarithm of provided loans. Source: National Bank of Slovakia 2018.

Relatively large shares of new housing loans are provided to repay older, usually more expensive loans. This is explained by the fact that new housing loans are not growing evenly along with the overall volume of housing loans.

Slovak households have significantly been using low interest rates and so the volume of housing loans did not decrease even during the financial crisis. The volume of new housing loans is currently almost ten times higher than in 2007, but the structure of subgroups of housing loans has changed significantly.

In 2007, the share of mortgage loans among total housing loans was 68.2% and the share of other real estate loans was 31.8%. In 2016 the share of mortgage loans decreased to 51.66% and the share of other real estate loans increased to 48.24%.

The issuing of mortgage bonds was lagging behind the provision of loans. This was related to less developed financial markets in the emerging economies and subsequently to the financial crisis, when the banks favoured other types of financing.

Since 2017, Slovak mortgage bonds are considered as covered bonds and the owners of these bonds are secured creditors. This improved the quality of Slovak covered bonds for investors. Slovak covered bonds "Krytý dlhopis" comply with the requirements of Article 52(4) UCITS (Undertakings Collective Investment in Transferable Securities) as well as of Article 129 of EU Regulation No. 575/2013. The listed covered bonds are eligible for repo transactions with the central bank (Páleníková et al. 2019).

The Table 4 shows significant changes in legislative and economic conditions and their impact on mortgage banking in Slovakia.

**Table 4.** An overview of the main changes in mortgage banking in Slovakia.

| Year | Change of Legislation or Economic Conditions | Specific Features |
|------|-----------------------------------------------|-------------------|
| **1996** | Amendment of the Act 21/1990 Coll. on banks | Defining the concepts of mortgage banking, conditions for entry into the sector, minimum required capital 1 billion Slovak crowns (after Euro adoption 33.3 Mil. EUR). |
| **1996** | Amendment of the Act 530/1990 Coll. on bonds | Conditions for issuing and covering of mortgage bonds; they could only be issued by a licensed bank, and the coverage was defined as regular and substitute. |
| **1999** | Amendment of the Act 21/1990 Coll. on banks | Introduction of state support for mortgage loans. |
| **2000** | Privatization of the largest banks in Slovakia | |
| **2002** | Adoption of a new Act 483/2001 Coll. on banks | New definition of mortgage and municipal loans; the possibility to provide loans also in foreign currency. |
| **2007** | Amendment of the Act 438/2001 Coll. on banks | Possibility of providing state support for loans to young people; |
| **2009** | Adoption of the Euro in Slovakia | |
| **2016** | Adoption of the Act 90/2016 Coll. on housing loans | Mortgage and municipal loans have become part of housing loans. |
| **2018** | Amendment of the Act 438/2001 Coll. on banks | Introduction of covered bonds instead of mortgage bonds as improvement of legislation closer to EU regulations. New requirements for LTV (Loan to value ratio), PTI (Payment to Income ratio) and DTI (Debt to Income ratio); LTV: 70%, PTI: the reserve must represent at least 30% of the client's net income, DTI: /ver must not be higher than eight times the client's annual income. Abolitishment of the obligation for a minimum capital of 33.3 Mil. EUR; abolishment of the obligation for a special license granted by National Bank of Slovakia for conducting mortgage business and its replacement by prior approval of National Bank of Slovakia (NBS); involvement of minimum requirements for overcollateralization and for disclosure; the possibility to include of hedging derivatives into the cover pool of covered bonds. |

Source: Processed on legislation sources; National Bank of Slovakia 2020.

The possibility of refinancing mortgage loans by another bank which has a different interest rate is used extensively in the Slovak Republic. This leads to strong competition among the banks and to efforts to retain clients, which is reflected in the decrease of interest rates for mortgage loans. The Figure 2 shows the continual decrease of the interest rates for mortgage loans fixed for different periods, especially after the onset of the financial crisis in 2008.

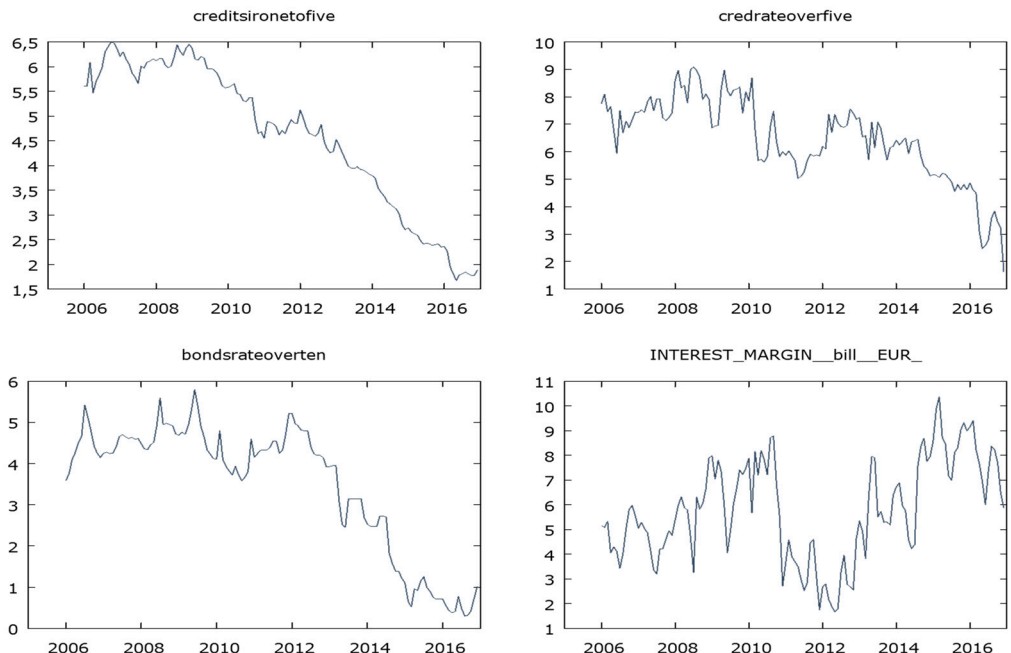

**Figure 2.** Interest rates on housing credits with maturity of one–five years (creditsironetofive), with maturity of over five years (credrateoverfive), mortgage bond rate with maturity of over ten years (bondsrateoverten) and interest margin (INTEREST_MARGIN) in the period 2006–2016.

The development of interest rates on mortgage loans with agreed maturity of over five years has had an overall decreasing trend; however, there is a higher rate of volatility. In 2016, the interest rates for less than five years and over five years came closer. For a period of ten years, the median interest rate from one to five years was 4.3% and the median interest rate over five years was 6.15%.

In the context of excessive indebtedness of Slovak households during the period of low interest, and with a certain increasing risk rate due to expected changes and other factors worsening the potential balance of income and household expenses, the National Bank of the Slovak Republic, as the regulator, introduced measures, the goal of which is to prevent negative effects of said changes, so households are capable of coping with the increasing financial burden.

The National Bank of Slovakia responded to the rising risks with recommendations for increased caution in the area of using mortgage and consumer loans on the part of clients as well as banks.

As part of the above changes in 2017 and 2018, three basic indicators were given a specific qualitative and quantitative definition. These were the Loan to Value Ratio (LTV), Payment to Income Ratio (PTI) and the Debt to Income Ratio (DTI). The main reason for tightening mortgages is the excessive growth of indebtedness of households, which could become a risk after a possible increase in interest rates. The indebtedness of Slovak households has been growing fastest since 2005 among the V4 countries and the ratio of household indebtedness growth to GDP is above average even within the Eurozone. It is clear that the availability of credit resources affects the prices of real estate. The sensitivity level differs based on the individual regions of Slovakia.

With improvement of covered bond legislation closer to EBA recommendations, the new legislation was adopted from 1 January 2018.

The main changes in legislation were: "1. Abolition of the special license granted by the National Bank of Slovakia for conducting mortgage business and its replacement by prior approval of National Bank of Slovakia; 2. Abolition of financing ratio; 3. stipulation of the minimum requirements for overcollateralization; 4. inclusion of hedging derivatives into the cover pool of covered bonds; 5. obligation of keeping a liquidity buffer; 6. obligatory stress testing; 7. special maturity extension of covered bonds in case of bankruptcy of the bank as issuer; 8. new requirements for disclosure." (Páleníková et al. 2019).

Following these changes in legislation, the environment for conducting mortgage transactions has changed significantly in Slovakia. The possibility of the regulator's influence on the environment of mortgage transactions has been significantly strengthened. On the credit side, this is through credit restrictions, and on the covered bond side this is through the approval of issues by regulators. This is a response to the need to increase market stability due to the rapid indebtedness of Slovak households.

The analysis of seasonality in the development of selected indicators points to the fact that seasonality has a major effect on taking, providing and issuing of new loans. The Figures 3–5 point out the high seasonality of mortgage banking indicators. The Figure 3 shows the seasonality of the overall provided mortgage loans and it points to an increased interest in mortgage loans in the months between May and October, which is logical given seasonality in the construction sector and interest in moving house during the summer months.

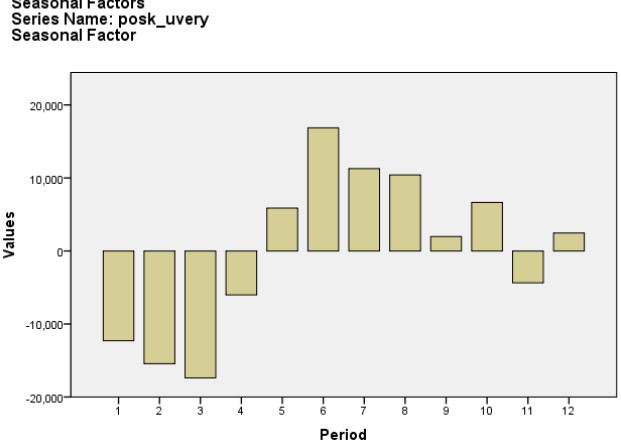

**Figure 3.** Seasonality is observed in provided credits.

Seasonality is also reflected in interest rates. As the Figure 4 shows, negative seasonal component of interest rates on mortgage loans with an agreed maturity of one–five years supported interest in mortgage loans in the months between April and July.

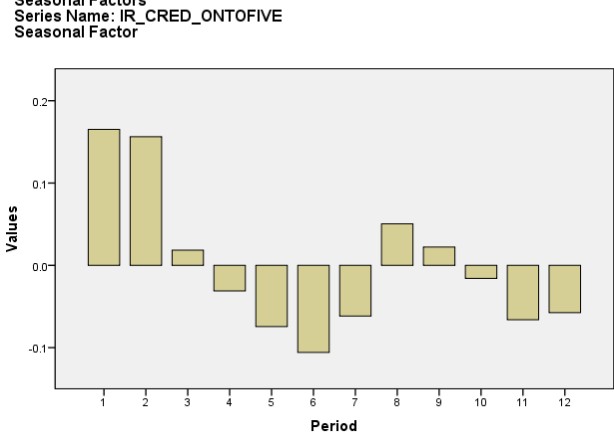

**Figure 4.** Seasonality of interest rates on mortgage loans with agreed maturity of one–five years.

As Figure 5 shows, seasonality of real estate prices points to the fact that real estate prices are growing the most during the summer time, which is linked to the willingness to move house as well as to other factors that describe the seasonality of the construction sector and behavior of people. During the analyzed period between 2006 and 2016 real estate prices have been growing in all regions of Slovakia.

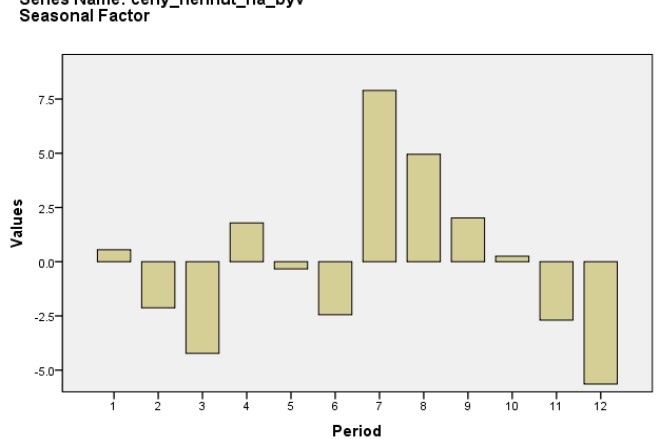

**Figure 5.** Seasonality of real estate prices.

Figure 6 shows the residential real estate prices in the regions of Slovakia with the highest prices—in Bratislava, Banska Bystrica, Zilina and Kosice during the period 2005–2018.

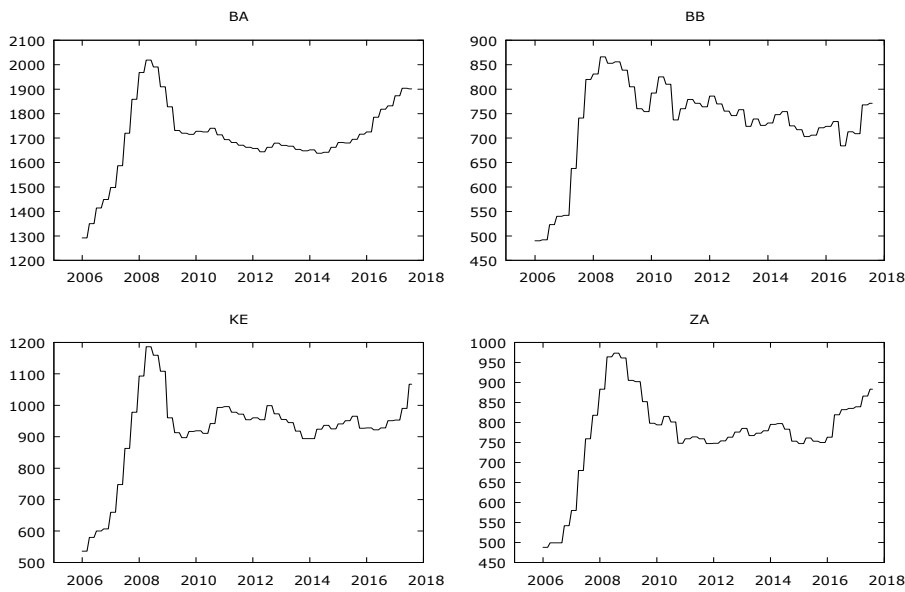

**Figure 6.** Residential real estate prices in the regions with the highest prices—Bratislava, Banska Bystrica, Zilina and Kosice in 2005–2018. Source: own processing using the sources of the National Bank of Slovakia 2018. BA = residential real estate prices in the Bratislava region; KE = residential real estate prices in the Kosice region; BB = residential real estate prices in the Banska Bystrica region; ZA = residential real estate prices in the Zilina region.

The prices in other regions in Slovakia were between the prices of the Kosice and Nitra regions. They grew the most in the Bratislava region and also reached the highest level in absolute terms. The most significant growth of real estate prices, except for the Bratislava region, was in the Zilina and the Kosice region. In 2018, real estate prices reached levels close to those of the financial crisis of 2008.

Figure 7 shows the residential real estate prices in the regions of Slovakia with lower prices—in Trnava, Nitra, Trencin and Presov during the period 2005–2018.

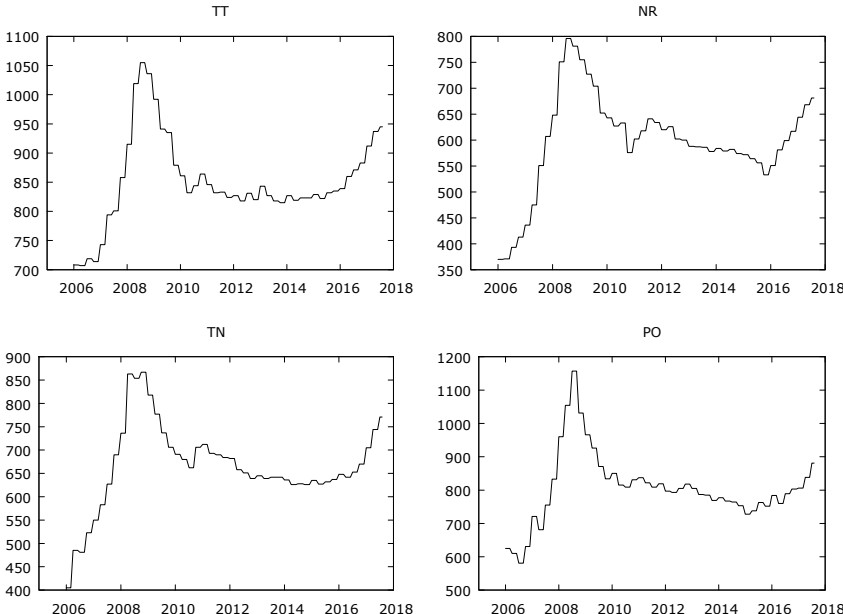

**Figure 7.** Residential real estate prices in the regions with lower prices—Trnava, Nitra, Trencin and Presov in 2005–2018. Source: Reproduced from National Bank of Slovakia 2018. TT = residential real estate prices in the Trnava region; NR = residential real estate prices in the Nitra region; TN = residential real estate prices in the Trencin region; PO = residential real estate prices in the Presov region.

## 5. Results

The aim of the paper is to evaluate the substantial links between basic economic indicators, housing finance indicators and real estate prices in Slovakia. To evaluate these issues, VaR models, models of panel regression and linear regression models were used. It was assumed that individual countries have their specific indicators of mortgage market development. The main part of the analysis focuses on the technical efficiency of individual banks, using DEA models.

### 5.1. Results of VAR Model for Slovak Real Estate Financing Indicators

Using the VAR model, the following impulse-response functions were analyzed. Appendix A shows the numerical results of the VaR models; the time lag of the variables is given in parentheses.

The following Figure 8 shows an impulse in the form of a logarithm of GDP (Gross Domestic Product) growth and a response in the form of a logarithm of new loans growth manifested within 1–2 months.

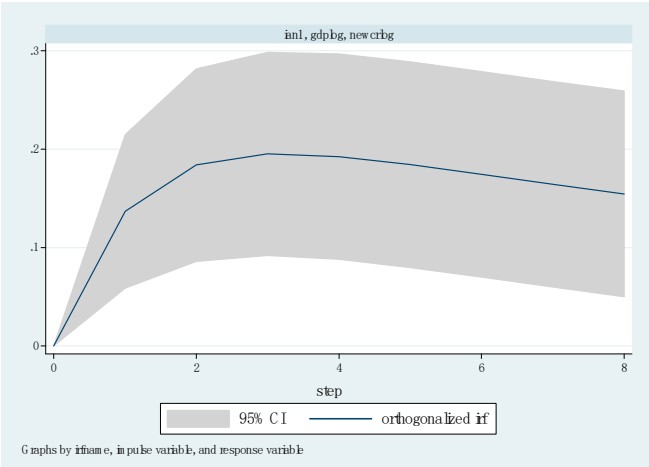

**Figure 8.** Shows the response of the new credits logarithm to the impulse—GDP logarithm.

Figure 9 shows an impulse in the form of a logarithm of salaries growth and a response in the form of a logarithm of new loans growth manifested within two months.

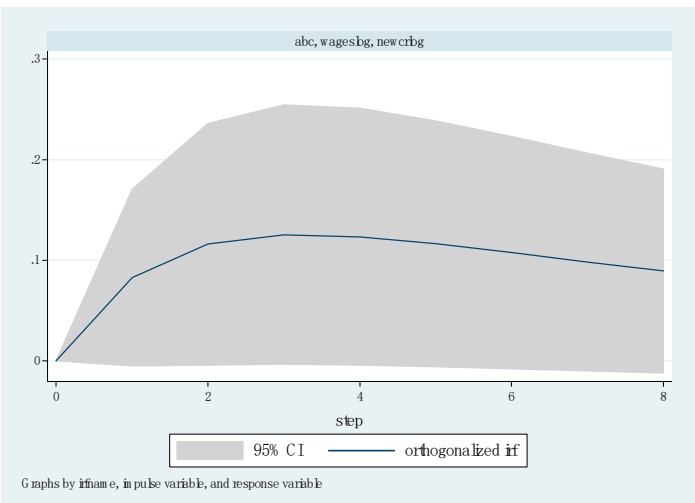

**Figure 9.** Impulse in the form of a logarithm of salaries growth and a response in the form of a logarithm of new loans growth.

Figure 10 shows an impulse in the form of growth of unemployment and a response in the form of a logarithm of decrease in new loans manifested within two months.

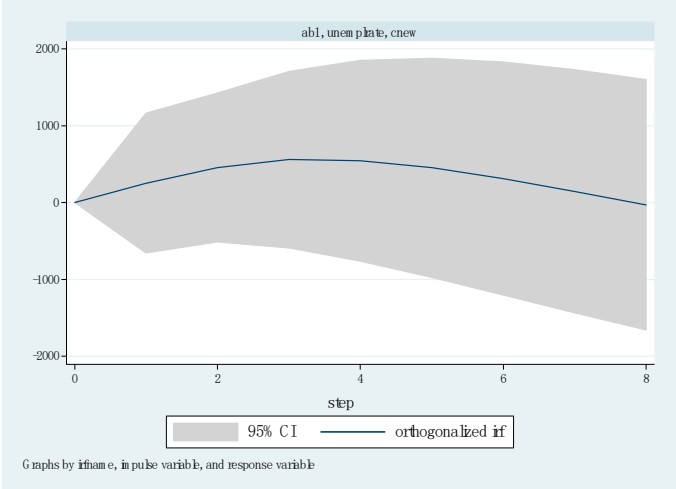

**Figure 10.** Shows an impulse in the form of growth of unemployment and a response in the form of a logarithm of decrease in new loans.

Figure 11 shows an impulse in the form of growth of loans taken and a response in the form of growth of real estate prices manifested within 15 months.

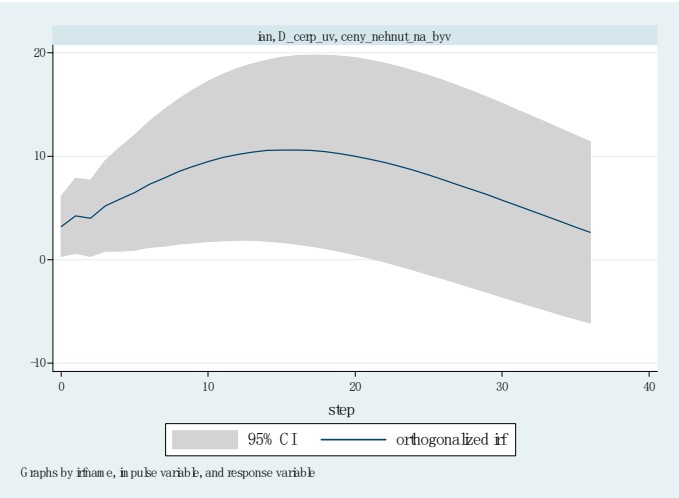

**Figure 11.** Impulse in the form of growth of loans and a response in the form of growth of real estate prices.

Figure 12 shows an impulse in the form of salary growth and a response in the form of growth of real estate prices in the Bratislava region manifested in the growth of real estate prices within 20 months.

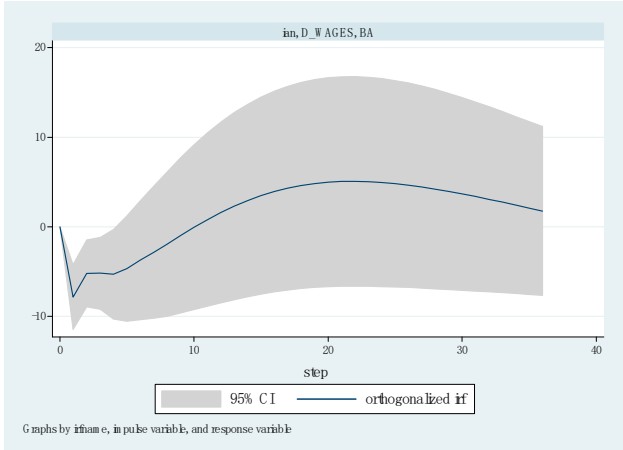

**Figure 12.** Impulse in the form of salary growth and a response in the form of growth of real estate prices in the Bratislava region.

### 5.2. Results of Regression Investigation

We were interested in finding out what the links are between the selected indicators of the mortgage market, and an analysis of the mortgage market based on panel regression models was carried out. Data used in panel regression models were identical to the data used in the DEA models. The analyzed period was 2008–2017. The panel regression models were estimated in the R-studio statistical system. The Table 5 contains model 1 and shows the relation between housing loans (HOUS_CRED) and indicators of banking market—mortgage bonds (MTG_BONDS), deposits (DEPOS), profit (PROFIT) and interest margin (IM).

**Table 5.** HOUS_CRED ~ MTG_BONDS + DEPOS + PROFIT + IM. (Balanced Panel: $n = 8$, $T = 9$, $N = 72$).

| Type of Panel Regression Model | Pooling | Fixed | Random | Between |
|---|---|---|---|---|
| **(Intercept)** | 40.1531 (9.7305) (4.1265) *** | - | 40.3957 (9.8767) (4.0900) *** | 60.8274 (34.2038) (1.7784) |
| **MTG_BONDS** | −0.1230 0.1143 −1.0766 | −0.0134 0.1334 −0.1005 | −0.1052 0.1166 −0.9017 | −0.4251 0.3155 −1.3474 |
| **DEPOS** | −0.2952 0.1201 −2.4586 * | −0.2942 0.1318 −2.2325 * | −0.2911 0.1209 −2.4080 * | −0.5881 0.4282 −1.3736 * |
| **IM** | 0.3257 0.1130 2.8 ** | 0.1053 0.1697 0.620 | 0.2994 0.1200 2.4941 * | |
| **R-Squared:** | 0.21363 | 0.10434 | 0.18607 | 0.8014 |
| **F-statistic:** | 4.55054 | 1.74741 | 3.82924 | 3.0265 |
| ***p*-value:** | 0.00259 | 0.15145 | 0.00731 | 0.1948 |

Hausman Test
data: Y ~ X
chisq = 3.7867, df = 4,
*p*-value = 0.4356
alternative hypothesis: one model is inconsistent

Source: Reproduced from National Bank of Slovakia 2018. Values in parentheses are std. error and t-value and *** denotes statistical significance at the 1% level, ** at the 5% level and * at the 10% level.

Model 1 shows the negative relationship between housing loans and deposits. The interpretation of this relationship suggests that the growing housing loans market had potential in other segments than in deposits, for example, in the change in assets structure in favor of housing loans. Housing loans represented an important means of placing resources for Slovak banks and especially during the financial crisis they represented a way of maintaining bank profits.

The volume of housing loans was in positive relation to the interest margin growth. We assume that increasing volume of housing loans contributed to this interest margin growth. The analysis of the development of housing loans for 2006–2017 points not only to their absolute growth, but also to the growth of their share within bank assets, which is supported by the results in model 1. Mortgage and housing loans from the point view of the banks are profitable in the initial period of their duration. This follows from the characteristics of annuity repayment. In the initial period, most of the repayment falls on interest and a smaller part on debt repayment. The growth of new loans contributes to the growth in the interest margin. During the financial crisis, they became an important tool for banks to deploy their resources. Although the interest rates on mortgage loans decreased, in relation to other investment alternatives the mortgage loans were a suitable combination of profits and risks for banks in the Slovak Republic.

The Table 6 contains panel regression model 2 which investigated the relationship between interest margin (IM) and factors as mortgage credits (MTG_CRED), other housing credits (OTHER_CRED), mortgage bonds (MTG_BONDS) and deposits (DEPOS).

**Table 6.** IM ~ MTG_CRED + OTHER_CRED + MTG_BONDS + DEPOS. (Balanced Panel: $n = 8$, $T = 9$, $N = 72$).

| Type of Panel Regression Model | Pooling | Fixed | Random | Between |
|---|---|---|---|---|
| (Intercept) | 35.6204 11.2505 3.1661 ** | - | 36.1165 9.7871 3.6902 *** | −241.468 117.5648 −2.0539 |
| MTG_CRED | 0.2712 0.1305 −2.0788 * | 0.0105 0.1122 0.0939 | −0.0683 0.1138 −0.6004 | 1.0552 0.8813 1.1973 |
| OTHER_CRED | 0.1533 0.1254 1.2226 | 0.0229 0.0928 0.2473 | 0.0566 0.0975 0.5803 | 5.3897 2.0572 2.6199 |
| MTG_BONDS | 0.2447 0.1365 1.7926 | 0.1570 0.1074 1.4615 | 0.1762 0.1117 1.5769 | 0.8332 0.5285 1.5766 |
| DEPOS | −0.1099 0.1301 −0.8452 | −0.1795 0.1073 −1.6727 | −0.1558 0.1102 −1.4131 | 0.3655 0.4906 0.7450 |
| R-Squared: | 0.1278 | 0.12886 | 0.10604 | 0.80835 |
| F-statistic: | 2.45426 | 2.21875 | 1.98676 | 3.16335 |
| *p*-value: | 0.054126 | 0.077595 | 0.10652 | 0.18563 |

Hausman Test

data: Y ~ X

chisq = 29.941, df = 4, $p$-value = $5.031 \times 10^{-6}$

alternative hypothesis: one model is inconsistent

Source: Reproduced from National Bank of Slovakia 2018. Values in parentheses are std. error and t-value and *** denotes statistical significance at the 1% level, ** at the 5% level and * at the 10% level.

Model 2 shows the positive relationship between interest margin and other real estate credits. Other real estate credits represent more risky loans with a higher LTV (Loan to Value) ratio and therefore the banks charged higher interest rates for this type of loan. Mortgage bonds were in a

positive relationship to the interest margin and low interest rates on quality bonds are assumed to have a beneficial effect on the interest margins of the banks.

Furthermore, the analysis of mortgage market indicators was performed on the basis of linear regression models listed in the Table 7.

**Table 7.** Linear regression models.

| | Model 1 | Model 2 | Model 3 | Model 4 | Model 5 |
|---|---|---|---|---|---|
| **Explanatory variables** | **Dependent Variables** | | | | |
| | **New Credits** | **Provided Credits** | **Interest Margin** | **Real Estate Prices** | **New Credits** |
| Constatnt | 737,626.70 (116,910.9) (6.31) *** | −712,904.093 (360,980.1) (−1.975) * | −15,885,977.7 (4,811,514) (−3.302) ** | −151.422 (24.870) (−6.089) *** | 227,338.55 (161,157.1) (1.411) |
| Real Estate Prices in Slovakia | −0.136 (83.794) (−3.658) *** | −0.487 (1321.185) (−5.962) *** | | | |
| Real Estate Prices in Bratislava | | 0.404 (1150.186) (4.671) *** | −0.284 (2495.484) (2.820) ** | 0.979 (0.013) (62.34) *** | 0.603 (416.150) (2.681) ** |
| Interest Rate on Credits 1–5 Years | 0.388 (9946.363) (−7.376) *** | -0.219 (31,083.719) (−9.571) *** | 0.229 (350,450.966) (2.161) * | 0.067 (1.811) (3.123) ** | |
| Interest Rates on Credits Over 5 Year | | | −0.348 (250,540.72) (−3.469) ** | 0.146 (1.088) (9.366) *** | 0.061 (7804.136) (1.226) |
| Deposits | −0.100 (0.003) (−1.870) * | 0.207 (0.013 (6.491) *** | 0.990 (0.117) (6.211) | −0.114 (0.001) (−4.594) *** | 0.024 (0.004) (0.349) |
| New Credits | | | −0.905 (2.594) (−4.674) *** | −0.034 (0.001) (−1.123) | |
| Long Term Interest Rates Debt Securities 10 Years Maturity | | | | | 0.395 (9851.10) (6.717) *** |
| Mortgage Bonds | 0.778 (0.17) (16.185) *** | 0.598 (0.056) (26.915) *** | 0.629 (0.845) (3.484) ** | 0.163 (0.001) (5.808) *** | 0.782 (0.017) (15.665) *** |
| R Square | 0.930 | 0.987 | 0.649 | 0.992 | 0.948 |
| Standard Error | 73,026.070 | 224,278.55 | 2,205,760.6 | 11.22 | 63,586.32 |
| F Statistic | 448.060 | 2084.917 | 40.994 | 2578.23 | 344.276 |
| Sig. | 0.000 | 0.000 | 0.000 | 0.000 | 0.000 |

Values in parentheses are std. error and t-value and *** denotes statistical significance at the 1% level, ** at the 5% level and * at the 10% level.

New loans are very important in mortgage banking. The specific method of annuity repayment results in the banks collecting the majority of the income from loan interests during the initial stages of their repayment. The annuity payment is a regular payment of the same amount, but with a changing ratio between the amounts intended for repaying the loan and debt amortization. Therefore, banks are

interested in new mortgage loans. Conversely, real estate loans in their final repayment stage are less attractive to the banks. We have therefore also included in the analysis models with new real estate loans.

Linear regression model 1 analyzed the period 2008–2016 and shows a positive relationship between new real estate loans (new credits) and the interest rates on loans with a maturity of one–five years. There is an inverse relation between real estate prices in Slovakia and new housing loans. From a national perspective, real estate price growth has slowed down new real estate loans. Deposits were also in an inverse relation to new loans. There was a significantly positive relationship between new loans and issued mortgage bonds.

Linear regression model 2 analyzes the relationship between volumes of all provided housing loans on the one side and real estate prices in Slovakia, real estate prices in Bratislava, interest rates and deposits on the other. This model points to an inverse relationship between real estate prices in Slovakia and the total volume of housing loans. Linear regression model 2 shows that this trend did not apply to real estate prices in Bratislava. The real estate prices grew together with the growing volume of loans. This suggests that real estate prices in Bratislava had a completely different development compared to the prices of real estate in other regions of Slovakia. The total volume of provided housing loans was in inverse relationship to the development of the interest rate on loans with maturity of one–five years and in a positive relationship to the development of deposits.

Linear regression model 3 analyzes the relationship between the interest margin of the banks and variables such as real estate prices, interest rates, deposits and new credits. There was a positive relationship between interest margin and interest rate on loans with maturity of one–five years and there was an inverse relationship between net interest margin on one side and interest rate for loans with maturity over five years on the other. This can be explained through customers' preferences to fix interest rates for five years during periods with higher interest rates, and during periods with low interest rates clients preferred fixing interest rates for periods longer than five years, which led to the decrease in the interest margin. The large growth of new loans during periods of low interest rates leads to a reduction of banks' margins. Overall, mortgage bonds contributed to the creation of interest margin for the banks.

Linear regression model 4 shows that the growth of real estate prices nationally was significantly driven by the growth of real estate prices in Bratislava. Overall, during the examined period, the price of real estate grew, together with growing interest rates on loans for real estate.

Linear regression model 5 shows the significant growth of real estate prices in Bratislava as a result of the growing number of provided credits. This model also confirms a positive relationship between the yields on long-term securities, mortgage bonds and the growth of new loans.

*5.3. Results Related to the DEA*

The analysis of the technical efficiency of Slovak banks with a mortgage business license points out the areas in which individual banks show specific characteristics. The list of banks included in the DEA models is in Appendix B.

A summary overview of the technical efficiency of mortgage banks in Slovakia points out their relative advantages and disadvantages. BCC models with variable returns were used for the analysis because they consider the size of the banks. Three output oriented models have been used, which gives recommendations to improve the output of banks, and one input oriented model, which gives recommendations to improve the efficiency of the banks' inputs.

The Table 8 shows the creation of DEA models: the inputs and outputs of the models, and the type of model used—input oriented or output oriented, respectively—with constant or variable returns to scale (CRS or VRS).

**Table 8.** Data Envelopment Analysis (DEA) Models of Technical Efficiency measurement.

| DEA Model No. | Input 1 | Input 2 | Output 1 | Orientation | Typ of Model | The Highest Average Efficiency of the Bank | The Highest Average Efficiency in the Year |
|---|---|---|---|---|---|---|---|
| 1. | Mortgage Credits | - | Margin | Output oriented | Banker-Charnes-Cooper (BCC) | Tatra banka | 2009 |
| 2. | Mortgage Credits | - | Profit | Output oriented | BCC | Tatra banka | 2007 |
| 3. | Mortgage Bonds | Deposits | Mortgage credits | Output oriented | BCC | SLSP Uni Credit | 2008 |
| 4. | Deposits | - | Mortgage credits | Input oriented | BCC | SLSP | 2008 |

Source: Reproduced from National Bank of Slovakia 2018.

The next Table 9 shows the results of the defined DEA models for banks in Slovakia.

**Table 9.** Results of DEA model 1 (Output oriented, VRS (Variable Returns to Scale) model; input Mortgage Credits, output Interest Margin).

| DMU */ | 2007 | 2008 | 2009 | 2010 | 2011 | 2012 | 2013 | 2014 | 2015 | 2016 | Average Efficiency of Bank |
|---|---|---|---|---|---|---|---|---|---|---|---|
| Bank 1 | 1 | 1 | 0.851 | 0.815 | 0.844 | 0.884 | 1 | 1 | 0.953 | 0.901 | 0.925 |
| Bank 2 | 1 | 1 | 1 | 1 | 1 | 1 | 1 | 1 | 1 | 1 | 1 |
| Bank 3 | 0.519 | 0.474 | 1 | 1 | 1 | 1 | 1 | 1 | 1 | 1 | 0.899 |
| Bank 4 | 0.474 | 0.987 | 1 | 1 | 1 | 1 | 1 | 1 | 1 | 0.954 | 0.888 |
| Bank 5 | 0.666 | 0.772 | 0.993 | 0.878 | 0.784 | 0.634 | 0.675 | 0.656 | 0.704 | 0.664 | 0.743 |
| Bank 6 | 0.442 | 0.371 | 0.592 | 1 | 0.637 | 0.632 | 0.712 | 1 | 1 | 1 | 0.738 |
| Bank 7 | 1 | 1 | 1 | 0.767 | 1 | 1 | 1 | 0.752 | 0.638 | 0.490 | 0.864 |
| Bank 8 | 1 | 0.685 | 0.750 | 0.722 | 0.775 | 0.700 | 0.782 | 0.783 | 0.156 | 0.157 | 0.651 |
| Average of the Year | 0.828 | 0.788 | 0.898 | 0.882 | 0.880 | 0.826 | 0.779 | 0.784 | 0.691 | 0.660 | 0.801 |

Source: Reproduced from National Bank of Slovakia 2018. */ DMU = Decision Making Unit.

The DEA method presents results on relative efficiency. The efficiency results of the banks were different if the set of banks or the selected inputs and outputs were changed. The interpretation of the results must take into consideration the presented restrictive conditions of the analysis.

DEA Model 1 shows that the most efficient bank within the analyzed group with mortgage credits as input and interest margin as output was Bank 2. From a time perspective, under said combination of inputs and outputs, the banks achieved their highest efficiency in 2009. We placed this result in the context of the high interest rates that peaked in 2009.

The Table 10 shows the results of DEA model 2.

**Table 10.** Results of DEA model 2 (output oriented, VRS model, input Mortgage Credits, output Profit).

| DMU | 2007 | 2008 | 2009 | 2010 | 2011 | 2012 | 2013 | 2014 | 2015 | 2016 | Average Efficiency of Bank |
|---|---|---|---|---|---|---|---|---|---|---|---|
| Bank 1 | 1 | 1 | 1 | 1 | 0.914 | 0.567 | 0.978 | 1 | 0.895 | 1 | 0.935 |
| Bank 2 | 1 | 1 | 1 | 1 | 1 | 1 | 1 | 1 | 1 | 0.854 | 0.985 |
| Bank 3 | 0.909 | 0.882 | 0.199 | 0.983 | 1 | 1 | 1 | 1 | 1 | 0.917 | 0.889 |
| Bank 4 | 0.540 | 0.363 | 0 | 0.786 | 0.827 | 0.596 | 0.816 | 0.680 | 0.748 | 0.560 | 0.592 |
| Bank 5 | 0.162 | 0.142 | 0 | 0 | 0.072 | 0 | 0.125 | 0 | 1 | 0 | 0.150 |
| Bank 6 | 1 | 0 | 0 | 0.173 | 0 | 0 | 0 | 0.102 | 0.349 | 0.254 | 0.187 |
| Bank 7 | 0.386 | 1 | 0 | 0.674 | 0 | 0 | 0.824 | 1 | 0 | 0 | 0.388 |
| Bank 8 | 1 | 1 | 1 | 0.638 | 1 | 0.506 | 0.110 | 0.134 | 0.136 | 0.106 | 0.563 |
| Average of the Year | 0.749 | 0.673 | 0.399 | 0.657 | 0.601 | 0.458 | 0.607 | 0.614 | 0.641 | 0.461 | 0.586 |

Source: Reproduced from National Bank of Slovakia 2018.

According to DEA model 2, the most efficient bank (within the analyzed group of banks) with mortgage credits as input and profit as output was the Bank 1. From a time perspective, under said combination of inputs and outputs, the banks achieved their highest efficiency in 2007. We link this result to the enormous growth of the credit market during the real estate price boom.

The Table 11 shows the results of DEA model 3.

**Table 11.** Results of DEA model 3 (output oriented, VRS, output Mortgage Credits, input: Mortgage Bonds and Deposits).

| DMU | 2008 | 2009 | 2010 | 2011 | 2012 | 2013 | 2014 | 2015 | 2016 | Average Efficiency of Bank |
|---|---|---|---|---|---|---|---|---|---|---|
| Bank 1 | 1 | 0.896 | 0.875 | 0.788 | 0.785 | 0.752 | 0.748 | 0.816 | 0.853 | 0.835 |
| Bank 2 | 0.693 | 0.562 | 0.690 | 0.639 | 0.669 | 0.651 | 0.729 | 0.699 | 0.691 | 0.669 |
| Bank 3 | 1 | 1 | 1 | 1 | 1 | 1 | 1 | 1 | 1 | 1 |
| Bank 4 | 0.663 | 0.593 | 0.569 | 0.546 | 0.643 | 0.595 | 0.648 | 1 | 1 | 0.695 |
| Bank 5 | 1 | 0.783 | 0.479 | 0.725 | 0.761 | 0.386 | 0.368 | 1 | 1 | 0.722 |
| Bank 6 | 1 | 0.526 | 0.912 | 1 | 1 | 0.305 | 0.478 | 0.620 | 0.780 | 0.736 |
| Bank 7 | 1 | 1 | 1 | 1 | 1 | 0.488 | 0.554 | 0.383 | 0.359 | 0.754 |
| Bank 8 | 1 | 1 | 1 | 1 | 1 | 1 | 1 | 1 | 0.961 | 0.998 |
| Average of the Year | 0.919 | 0.795 | 0.815 | 0.837 | 0.857 | 0.647 | 0.690 | 0.815 | 0.835 | 0.801 |

Source: Reproduced from National Bank of Slovakia 2018.

Result of DEA model 3 shows that the most efficient bank within the analyzed group of banks with mortgage credits as input and the mortgage bonds and deposits as outputs was Bank 3. From a time perspective, under said combination of inputs and outputs, the banks achieved their highest efficiency in 2008. We put this result in the context of the enormous growth of the credit market and the ability to effectively allocate resources into mortgage loans.

The Table 12 shows the results of DEA model 4.

**Table 12.** Results of DEA model 4 (Input oriented, VRS, input Deposits, and output Mortgage Credits).

| DMU | 2008 | 2009 | 2010 | 2011 | 2012 | 2013 | 2014 | 2015 | 2016 | Average Efficiency of Bank |
|---|---|---|---|---|---|---|---|---|---|---|
| Bank 1 | 1 | 0.889 | 0.768 | 0.671 | 0.645 | 0.519 | 0.674 | 0.748 | 0.790 | 0.744 |
| Bank 2 | 0.597 | 0.502 | 0.697 | 0.640 | 0.666 | 0.512 | 0.581 | 0.548 | 0.446 | 0.576 |
| Bank 3 | 0.987 | 1 | 1 | 1 | 1 | 1 | 1 | 1 | 1 | 0.998 |
| Bank 4 | 0.605 | 0.611 | 0.583 | 0.547 | 0.558 | 0.353 | 0.450 | 1 | 1 | 0.634 |
| Bank 5 | 1 | 0.818 | 0.662 | 0.728 | 0.639 | 0.000 | 0.000 | 0.295 | 0.314 | 0.495 |
| Bank 6 | 0.723 | 0.612 | 0.730 | 0.576 | 0.540 | 0.000 | 0.000 | 0.216 | 0.199 | 0.399 |
| Bank 7 | 1 | 1 | 1 | 1 | 1 | 0.000 | 0.000 | 0.253 | 0.287 | 0.615 |
| Bank 8 | 0.358 | 0.403 | 0.353 | 0.387 | 0.359 | 1 | 1 | 1 | 1 | 0.651 |
| Average of the Year | 0.784 | 0.729 | 0.724 | 0.693 | 0.676 | 0.423 | 0.463 | 0.632 | 0.629 | 0.6392 |

Source: Reproduced from National Bank of Slovakia 2018.

Model 4 shows that the most efficient bank within the analyzed group of banks with deposits as input and mortgage credits as output was Bank 3. From a time perspective, under said combination of inputs and outputs, the banks achieved their highest efficiency in 2008. We put this result in the context of the enormous growth of the credit market and the ability to effectively allocate resources into mortgage loans.

These results of DEA analysis are consistent with those of Anayiotos et al. (2010), who found that banks showed their highest efficiency in the pre-crisis year of 2007. Domestic macroeconomic developments were extremely favorable in 2007. Economic growth was at a record high level due to the positive contribution of domestic and foreign demand. By sector, corporate loans went mainly to trade, industrial production and construction of real estate. Housing loans predominated in retail loans

(National Bank of Slovakia 2007). In the period after the financial crisis, Slovakian banks' performance in mortgage banking did not reach pre-crisis levels.

Our results for V4 were similar to those of Pancurova and Lyocsa (2013), who used the mediation approach. The results for the period 2005–2008 found the banking sector in the Czech Republic the most efficient.

Sunega and Lux (2007), on the Czech mortgage market state that "the high degree of concentration does not have to be necessarily a sign of inefficiency; and that . . . the relatively high degree of competitiveness of mortgage lenders (proved by low and decreasing margins, growing product complexity and increased both maximum and average LTV) is employed only on recruitment of new clients", and this can also be stated about banks in Slovakia in the comparable period.

## 6. Conclusions

The stabilization of macroeconomic indicators of the Slovakian economy, especially the lowering of interest rates, growth of employment and performance of the economy significantly contributed to the development of mortgage banking in Slovakia. Together with other factors, such as the administrative burden of mortgage coverage and cheap money, this caused a reduction in the volume of issued bonds.

Possible improvements in the quality of legislation could have a positive effect on the development of the mortgage bonds market. The condition is that the banks do not leave real mortgage loans for other housing loans that serve to cover the mortgage bonds.

Following the growth of housing loans combined with seasonality and the expected general growth of interest rates, the time has come for a more cautious approach to lending. A cautious approach can be seen in all of the monitored areas of housing loans: Loan to Value (LTV), Payment to Income Ratio (PTI), and sufficient financial reserves of consumers.

The development of mortgage banking introduces a positive synergy effect for economic entities. The mortgage banking system is capable of concentrating long-term resources and guarantees their providers adequate security and appreciation of resources invested in mortgage bonds.

The mortgage bonds system has a stabilizing effect on the economy and therefore it is necessary to continue support for positive changes leading to an increase of trust of the financial entities towards mortgage bonds.

In Slovakia, mortgage banking is focused more on providing mortgage loans, whereby the potential of mortgage bonds as financial tools suitable for investing and generating long-term resources is used to a lesser extent. In this context it is positive that Slovak legislation shifted the focus on regulation towards mortgage bonds, which are considered as covered bonds and thus Slovakia became one of the countries with a well-developed approach to the mortgage system and to investor protection.

There has been a significant increase of real estate prices over the past five years, especially in the Bratislava region. Previous analyzes confirmed that the availability of real estate loans had the greatest impact on this increase in prices.

Real estate prices in Bratislava therefore have significantly different development factors than real estate prices from a nationwide perspective. These issues will be the subject of future research. We assume that multiple ownership and real estate purchase for commercial purposes (for rent) play a role here, and therefore real estate prices are determined by other factors.

The seasonality of individual factors relating to mortgage financing confirmed that most loans are drawn during the summer months. The seasonality of the development of interest rates points to the fact that the banks react to the growing volume of loans by competing clients, which is reflected in the decrease in interest rates. Real estate prices increase during the periods of highest drawdown of loans.

The growth of new loans contributes significantly to the growth of interest margins. During the financial crisis, new housing credits became an important possibility for banks in which to invest their resources. Although the interest rates on mortgage loans decreased, in relation to other investment alternatives mortgage loans were an interesting combination of profits and risks for the banks in Slovakia.

These results of DEA analysis confirmed that banks showed their highest efficiency in the pre-crisis year of 2007. The three largest banks in Slovakia achieve the highest efficiency. Economic growth was at a record high level due to the positive contribution of foreign and domestic demand. A large part of domestic demand was represented by demand for real estate property.

The financial crises turned real estate financing into a safe form of investment for the developers and, for the banks, a suitable area for lending.

In mortgage banking, the interest rate has a dual role. On the one hand, the interest rate determines the yield for the bank. On the other hand, it serves as a yield criterion for determining the value of real assets (yield-angled yield criterion). However, from the bank's perspective, low interest rates can contain two types of risks: (1) low income from mortgage rate loans; and (2) low interest rate, as a factor of the valuation of real assets, can cause overvaluation of real estate as collateral.

If interest rates were to rise, the increase of incomes from loans would be reflected in the banks' profits only after a certain time, but the decrease of real estate valuation would be a serious risk.

From the client's perspective risk increases when extensive debt complicates the repayment of loans (in the case of a financial crisis and increase of unemployment). It will be necessary to deal with the factor of whether the client acquires the first or a subsequent property, i.e., whether it is intended for residence or business.

Therefore, it would be suitable for policymakers to react not only to the household indebtedness rate, but also to the development of interest rates in the banking market. Policy makers should not only see risks for individual clients due to their over-indebtedness, but should also see risks for banks, for possible changes in the real estate market, or for changes in interest rates in the future.

**Funding:** The APC was funded by Faculty of National Economy University of Economics in Bratislava. This research was funded by Faculty of National Economy University of Economics in Bratislava and in the framework of the project MUNI/A/1081/2019 too. All support is greatly acknowledged.

**Acknowledgments:** I would like to thank the two anonymous reviewers; they gave me valuable advice, suggestions and comments that have improved this article so that I can contribute to the interesting debate on real estate financing.

**Conflicts of Interest:** The author declares no conflict of interest. Author declares any personal circumstances or interest that may be perceived as inappropriately influencing the representation or interpretation of reported research results. Any role of the funders in the design of the study; in the collection, analyses or interpretation of data; in the writing of the manuscript, or in the decision to publish the results is declared.

## Appendix A

*Results of VAR Models*

D_GDP = 0.950561*D_GDP(−1) − 0.0365164*D_GDP(-2) − 0.0456925*D_WAGES(-1) − 0.1416053*D_WGES(-2) − 0.0310661*IR_CRED_ONTOFIVE(-1) + 0.0362129* IR_CRED_ONTOFIVE(-2) + 0.0081824* IR_CRED_OVERFIVE(-1) − 0.0201425* IR_CRED_OVERFIVE(-2) + 0.0081767*NEW_CRED(-1) + 0.0079901*NEW_CRED(-2) + 0.1010144;

D_WAGES = −0.0059308*D_GDP(-1) − 0.0007835*D_GDP(-2) + 0.9285374*D_WAGES(-1) − 0.0696951*D_WAGES(-2) + 0.0037041*IR_CRED_ONTOFIVE(-1) − 0.0060402* IR_CRED_ONTOFIVE(-2) + 0.0012907* IR_CRED_OVERFIVE(-1) − 0.0026857* IR_CRED_OVERFIVE(-2) + 0.0011656 *NEW_CRED(-1) + 0.0048618*NEW_CRED(-2) + 0.0479811;

IR_CRED_ONTO~E = −0.0910157 *D_GDP(-1) + 0.227333*D_GDP(-2) − 0.3389776*D_WAGES(-1) + 0.3707243*D_WAGES(-2) + 0.9813595*IR_CRED_ONTOFIVE(-1) + 0.0190953* IR_CRED_ONTOFIVE(-2) − 0.0557925* IR_CRED_OVERFIVE(-1) + 0.0232009* IR_CRED_OVERFIVE(-2) + 0.004056*NEW_CRED(-1) + 0.0010981*NEW_CRED(-2) + 0.1283714;

IR_CRED_OVER~E = +0.5206101*D_GDP(-1) − 0.2337225*D_GDP(-2) −
4.512529*D_WAGES(-1) + 4.059946*D_WAGES(-2) +
0.02550724*IR_CRED_ONTOFIVE(-1) − 0.0641126* IR_CRED_ONTOFIVE(-2) +
0.7006075* IR_CRED_OVERFIVE(-1) + 0.1728778* IR_CRED_OVERFIVE(-2) −
0.0777941*NEW_CRED(-1) − 0.0312844*NEW_CRED(-2) + 0.038906;

NEW_CRED = 0.8512959*D_GDP(-1) + 0.3333476*D_GDP(-2) + 2.160683*D_WAGES(-1) +
2.390888*D_WAGES(-2) − 0.708711*IR_CRED_ONTOFIVE(-1) + 0.9668978*
IR_CRED_ONTOFIVE(-2) − 0.0414462* IR_CRED_OVERFIVE(-1) + 0.1563089*
IR_CRED_OVERFIVE(-2) − 0.0076939*NEW_CRED(-1) + 0.1725653*NEW_CRED(-2)
−2.350028;

D_DRAW_CRED = 0.3767037*D_DRAW_CRED(-1) + 0.3904603*D_DRAW_CRED(-2) −
0.0356852*NEW_CRED(-1) − 0.003576* NEW_CRED(-2) −
0.0084384*REAL_EST_PRICES(-1) + 0.0010563*REAL_EST_PRICES(-1) +
0.0062992*REAL_EST_BA(-1) − 0.0004893*REAL_EST_BA(-2) + 0.3350388*D_GDP(-1) −
0.0456893*D_GDP(-2) − 0.6885124*D_WAGES(-1) + 0.7797594* D_WAGES(-2) +
0.2567508*IR_CRED_ONTOFIVE(-1) − 0.2082726* IR_CRED_ONTOFIVE(-2) +
0.0960518*IR_CRED_OVERFIVE(-1) + 0.0222058*IR_CRED_OVERFIVE(-2) − 1.412437;

NEW_CRED = 0.0006868*D_DRAW_CRED(-1) + 0.5441694*D_DRAW_CRED(-2) −
0.1256487*NEW_CRED(-1) + 0.0131956*NEW_CRED(-2) +
0.0029129*REAL_EST_PRICES(-1) − 0.0189832*REAL_EST_PRICES(-2) −
0.0055459*REAL_EST_BA(-1) + 0.0177095*REAL_EST_BA(-2) + 0.2311897*D_GDP(-1) −
0.2784483*D_GDP(-2) + 5,218712*D_WAGES(-1) + 0.7132276* D_WAGES(-2)
-0.6203518*IR_CRED_ONTOFIVE(-1) + 1.039223*IR_CRED_ONTOFIVE(-2) +
0.0610553*IR_CRED_OVERFIVE(-1) + 0.2359283 *IR_CRED_OVERFIVE(-2) − 4.8865;

REAL_EST_PRICES = + 5.731917*D_DRAW_CRED(-1) − 3.513364*D_DRAW_CRED(-2) −
0.9073634*NEW_CRED(-1) + 0.8512757*NEW_CRED(-2) +
0.4997239*REAL_EST_PRICES(-1) − 0.0377314*REAL_EST_PRICES(-2) +
0.221934*REAL_EST_BA(-1) + 0.1872306*REAL_EST_BA(-2) + 21.05149*D_GDP(-1)
-10.97258*D_GDP(-2) -263.4146*D_WAGES(-1) + 239.5108* D_WAGES(-2) +
2.480429*IR_CRED_ONTOFIVE(-1) + 2.157179*IR_CRED_ONTOFIVE(-2) +
0.0610553*IR_CRED_OVERFIVE(-1) + 0.2359283 *IR_CRED_OVERFIVE(-2) −54.37944;

## Appendix B

The banks in the sample of DEA models were the banks with special license granted by National Bank of Slovakia for conducting mortgage business:

| Bank Name: |
| --- |
| VUB, a. s. (General Credit Bank) |
| Tatra banka, a. s. (Tatra Bank) |
| SLSP, a. s. (Slovak Savings Bank) |
| CSOB, a. s. (Czechoslovakian Trade Bank) |
| OTP Banka Slovensko, a. s. (OTP Bank Slovakia) |
| Prima banka, a. s. (Prima Bank Slovakia; previously Dexia) |
| Sberbank, a. s. (Sberbank Slovakia, previously Volksbank) |
| UniCredit Bank Czech Republic and Slovakia, a.s. (UniCredit Bank) |

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
