# Peer review of "Twenty Years of Mortgage Banking in Slovakia"

_ijfs, doi:10.3390/ijfs8030056_

Round 1

Reviewer 1 Report

The paper empirically investigates the mortgage banking system development in Slovakia for the period 2006-2017, in order to evaluate the links between the economic indicators, housing finance and real estate prices using VaR, panel and linear regression and DEA.

Please verify statements like:

Row 18: “It can be assumed that Slovakia has the country specific indicators of mortgage banking development.”

Please explain abbreviation for DEA at first use and not in row 700, in title of 4.3.

Please revise; in my opinion, there is a contradiction between the two statements:

Rows 757-758: 757 … an increase of interest rates could have a positive effect on the development of the mortgage bonds market.

Rows 796-800 However, from the bank’s perspective, low interest rates can contain two types of risks – 1) low appreciation rate of bank loans and 2) low interest rate as a factor of valuation of real assets can cause over appreciation of real estates as collaterals. In case of the interest rates increasing in the future, the increase of the loans prices would be reflected in the banks’ profits only after a certain time, but the decrease of appreciation based on real estates would represent a serious risk.

Please develop the paragraph of Conclusions and present the limits of research and future research directions

In my opinion, the separation between 2. Methodology and data and 4. Results with section 3. The Development of Mortgage Banking in Slovakia in between, is artificial. I suggest The Development of Mortgage Banking in Slovakia to be placed before Methodology and data

I would suggest to extend explanations like: row 793-794:

“These results of DEA analysis confirmed that banks have the highest efficiency in the pre-crisis year of 2007. The first three largest banks in Slovakia achieve the highest efficiency.” Why?

Comma is not the decimal indicator in English; it should be replaced with point.

See for example Tables 4-7. Somehow, this is surprising, because in the first part of the paper it is used correctly.

I believe some references to other studies about the same topic in Europe might be useful, especially the study investigating the Czech Republic:

  1. Brissimis, S. N., & Vlassopoulos, T. (2009). The Interaction between Mortgage Financing and Housing Prices in Greece. The Journal of Real Estate Finance and Economics, 39(2), 146–164. https://doi.org/10.1007/s11146-008-9109-3
  2. Sunega, P., & Lux, M. (2007). Market-Based Housing Finance Efficiency in the Czech Republic. European Journal of Housing Policy, 7(3), 241–273. https://doi.org/10.1080/14616710701477888

Author Response

Dear reviewer,

Thank you very much for your valuable comments and advice, I agree with them and try to accept them. Your comments are inspiring for me and at the same time quite sensitively, for which I thank you especially.

My answers I mentioned directly in the text of review by individual comments.

Kind regards

Eva Horvátová 

Comments and Suggestions for Authors

The paper empirically investigates the mortgage banking system development in Slovakia for the period 2006-2017, in order to evaluate the links between the economic indicators, housing finance and real estate prices using VaR, panel and linear regression and DEA.

Please verify statements like:

Row 18: “It can be assumed that Slovakia has the country specific indicators of mortgage banking development.”

            The text has been changed as follows: "Slovakia has specific indicators of the development of mortgage banking, adequate to its historical and economic development".

Please explain abbreviation for DEA at first use and not in row 700, in title of 4.3.

               The abbreviation for DEA was inserted into the row 341: (Data Envelopment Analysis) and it was removed from the title of 4.3.

Please revise; in my opinion, there is a contradiction between the two statements:

Rows 757-758: 757 … an increase of interest rates could have a positive effect on the development of the mortgage bonds market.

            I agree that the positive effect of rising interest rates is limited and debatable, so I deleted the text.

Rows 796-800 However, from the bank’s perspective, low interest rates can contain two types of risks – 1) low appreciation rate of bank loans and 2) low interest rate as a factor of valuation of real assets can cause over appreciation of real estates as collaterals. In case of the interest rates increasing in the future, the increase of the loans prices would be reflected in the banks’ profits only after a certain time, but the decrease of appreciation based on real estates would represent a serious risk.

The text has been changed as follows:

In mortgage banking, the interest rate has a dual role. On the one hand, the interest rate determines the yield for the bank. On the other hand, the interest rate serves as a yield criterion for determining the value of real assets (yield-angled yield criterion). However, from the bank’s perspective, low interest rates can contain two types of risks – 1) low income from mortgage loans and 2) low interest rate as a factor of overvaluation of real estates as collaterals.

Please develop the paragraph of Conclusions and present the limits of research and future research directions

 The following text was added:

It will be necessary to deal with whether the client acquires the first or next residential property, i.e. whether it is intended for consumption or business.

In my opinion, the separation between 2. Methodology and data and 4. Results with section 3. The Development of Mortgage Banking in Slovakia in between, is artificial. I suggest The Development of Mortgage Banking in Slovakia to be placed before Methodology and data

 The first part before Methodology and Data was named as “Theoretical and Methodological Basis of Mortgage Market Research”; and the part after Methodology and Data was named as “The Mortgage Banking in Slovakia - Stylized Facts”.

I would suggest to extend explanations like: row 793-794:

“These results of DEA analysis confirmed that banks have the highest efficiency in the pre-crisis year of 2007. The first three largest banks in Slovakia achieve the highest efficiency.” Why?

 (819-821): Based on the annual report of the National Bank of Slovakia, the following text was added: Economic growth was at a record high level due to the positive contribution of foreign and domestic demand. A large part of domestic demand was represented by demand for real estate property.   

(760-764): Domestic macroeconomic developments were in 2007 extremely favorable. Economic growth was at a record high level due to the positive contribution of domestic and foreign demand.  By sector, corporate loans went mainly to trade, industrial production and construction of real estate. Housing loans predominated in retail loans (NBS, 2007).

Comma is not the decimal indicator in English; it should be replaced with point.

See for example Tables 4-7. Somehow, this is surprising, because in the first part of the paper it is used correctly.

I have been working on this article for more than a year. I prepared the tables in different periods and therefore they are formatted differently. Thank you for pointing out the incorrect decimal point.

I believe some references to other studies about the same topic in Europe might be useful, especially the study investigating the Czech Republic:

  1. Brissimis, S. N., & Vlassopoulos, T. (2009). The Interaction between Mortgage Financing and Housing Prices in Greece. The Journal of Real Estate Finance and Economics39(2), 146–164. https://doi.org/10.1007/s11146-008-9109-3
  2. Sunega, P., & Lux, M. (2007). Market-Based Housing Finance Efficiency in the Czech Republic. European Journal of Housing Policy7(3), 241–273. DOI: 10.1080/14616710701477888. 

You are very right. I know that Peter Sunega and Martin Lux are doing very good research on housing in the Czech Republic. They inspired me to work on the availability of housing. I would like to do that in the future. I am glad that you have brought to my attention the authors Brissimis, S. N., and Vlassopoulos. I will continue to follow their research.

The following texts have been added:

(row 770-775): Sunega and Lux (2007), on the sample of Czech mortgage market state that „the high degree of concentration does not have to be necessarily a sign of inefficiency; and that ... the relatively high degree of competitiveness of mortgage lenders (proved by low and decreasing margins, growing product complexity and increased both maximum and average LTV) is employed only on recruitment of new clients“- and this can also be stated about banks in Slovakia in a comparable period.

(247-254): An interesting methodological approach to the analysis of the real estate market was chosen by the authors Brissimis and Vlassopoulos (2009). They also involved in the analysis the time aspect: short-term and long-term view. The aim of their paper has been the analysis of the interaction between housing loans and housing prices in Greece using multivariate cointegration techniques. The results of their long-run analysis indicated that „housing prices are weakly exogenous, hence they do not react to disequilibria in the mortgage lending market. This suggests that in the long run a line of causality running from housing loans to housing prices is not confirmed.“ The short-run analysis indicated a bi-directional dependence among housing loans and housing prices.

(129-132): According to Sunega and Lux (2007), several authors agree that „the differences arise from local housing finance traditions, macro-economic performance, accessibility of funds, sources of capital for loan extensions, variety and types of mortgage products, variety of interest rate fixing, level and content of state interventions“.

Reviewer 2 Report

Thank You for the opportunity to review this interesting paper. 

The paper wis well written and quite professional. 

The only minor improvements I suggest:

  • develope the results discussion in relation to previous foreign studies to compare Slovakian situation with other countries

Author Response

Dear reviewer,

Thank you very much for the positive feedback on my article.
I am pleased with your trust.

Kind regards

Eva Horvátová